# Comparison of post-discharge mortality and medical expenditures in COVID-19 patients according to mechanical ventilation and extracorporeal membrane oxygenation use: The LIFE study

Jun Kawabata[1,2], Kenichi Goto ORCID[1]*, Megumi Maeda[3], Haruhisa Fukuda[3]

**1** Department of Health Sciences, Graduate School of Medical Sciences, Kyushu University, Fukuoka City, Fukuoka, Japan, **2** Advanced Critical Care Center, Kurume University Hospital, Kurume City, Fukuoka, Japan, **3** Department of Health Care Administration and Management, Graduate School of Medical Sciences, Kyushu University, Fukuoka City, Fukuoka, Japan

* goto.kenichi.638@m.kyushu-u.ac.jp

## Abstract

Outcomes related to health status and economic burden among patients who experienced critical COVID-19 remain insufficiently studied. We examined 180-day post-discharge mortality and total medical expenditures in COVID-19 patients according to their use of mechanical ventilation (MV) or extracorporeal membrane oxygenation (ECMO) during hospitalization. Using medical claims data from a Japanese municipality, this retrospective cohort study analyzed hospitalized COVID-19 patients who were discharged between April 1, 2020 and September 30, 2021. Patients were categorized into an MV/ECMO group (indicating severe disease) or a non-MV/ECMO group. Their differences in mortality and expenditures were compared using the χ² test and Mann–Whitney *U* test, respectively. A Cox regression analysis was performed to calculate the hazard ratios of MV/ECMO use for mortality, and a generalized linear model with gamma distribution was constructed to examine the association between MV/ECMO use and expenditures. The covariates included age, sex, comorbidities, and length of stay. The MV/ECMO group had significantly higher mortality (16.0% vs. 11.1%, *p* = 0.002) and expenditures ($8,732 vs. $3,460, *p* < 0.001) than the non-MV/ECMO group. MV/ECMO use was significantly associated with higher mortality (hazard ratio: 1.66, 95% confidence interval: 1.27–2.15); other risk factors included age (1.06, 1.05–1.07), dementia (1.48, 1.10–1.99), and cancer (1.92, 1.56–2.36). MV/ECMO use was also significantly associated with higher expenditures (Exp[β]: 1.49, 95% confidence interval: 1.29–1.73); other risk factors included kidney disease (1.60, 1.29–2.01), cerebrovascular disease (1.74, 1.56–1.94), and cancer (1.28, 1.14–1.44). Survivors of severe COVID-19 who required

which permits unrestricted use, distribution, and reproduction in any medium, provided the original author and source are credited.

**Data availability statement:** The data used in this study were acquired under agreements between Kyushu University and the participating municipalities, which stipulate that the data can only be used by authorized research institutions and cannot be shared with third parties. However, research institutions that have entered into agreements with the authorized research group in Kyushu University may access the data. Please contact the Joint Research Department of Kyushu University (ijkkyoudou@jimu.kyushu-u.ac.jp) for inquiries regarding data access.

**Funding:** ・ Initials of the authors who received each award →H.F ・ Grant numbers awarded to each author →JPMJFR205J ・ The full name of each funder →Japan Science and Technology Agency's FOREST (Fusion Oriented REsearch for disruptive Science and Technology) ・ URL of each funder website →https://www.jst.go.jp/souhatsu/en/index.html ・ Did the sponsors or funders play any role in the study design, data collection and analysis, decision to publish, or preparation of the manuscript? →No.

**Competing interests:** The authors have declared that no competing interests exist.

MV or ECMO during hospitalization were associated with higher post-discharge mortality and expenditures, suggesting a need for targeted care to reduce their clinical and economic burden.

## Introduction

The COVID-19 pandemic led to a surge in critically ill patients requiring intensive care worldwide [1]. Patients discharged from intensive care units (ICUs) after severe COVID-19 often experience persistent physical, mental, and cognitive impairments [2,3], and face increased risks of poor prognosis and higher healthcare costs post-discharge [4,5].

Previous studies have reported on the risk factors of mortality in individuals with severe COVID-19. For example, a nationwide cohort study in Sweden found that ICU-admitted COVID-19 patients who required mechanical ventilation (MV) had higher 360-day mortality than non-MV users, and that mortality was associated with sex and comorbidity burden [6]. In addition, a Japanese study found that age had a significant impact on in-hospital mortality among COVID-19 patients who required MV [7]. Furthermore, extracorporeal membrane oxygenation (ECMO) is used for patients with severe respiratory failure who do not respond to conventional mechanical ventilation. Previous studies have reported high in-hospital mortality and substantial healthcare resource use among COVID-19 patients requiring ECMO [8], highlighting the importance of evaluating their long-term outcomes after hospital discharge. At present, there is insufficient evidence on the factors influencing post-discharge mortality in severe COVID-19 patients in Japan.

Survivors of severe COVID-19 may also experience a heavier economic burden. A US study reported significant increases in healthcare utilization and expenditures in COVID-19 patients for 6 months after diagnosis [9]. Similarly, an Israeli study found that patients with long COVID had 2.5-times higher medical expenditures than patients without long COVID [10]. These findings suggest that COVID-19 patients, especially those with severe disease, may require medical care for a long period after completing acute treatment. While several factors influencing medical expenditures in COVID-19 patients have been identified [11], little is known about the determinants of long-term post-discharge medical expenditures in critically ill patients.

Identifying the factors associated with post-discharge outcomes may contribute to the implementation of targeted interventions to improve survival and reduce expenditures in severe COVID-19 patients. We hypothesized that the use of MV or ECMO, indicating severe acute disease, would increase post-discharge mortality and medical expenditures. To test this hypothesis, we examined the 180-day post-discharge mortality and total medical expenditures of hospitalized COVID-19 patients according to their use of MV or ECMO. We also sought to identify the risk factors (such as comorbidities) for the two outcomes.

## Methods

### Study design and data source

This retrospective cohort study used data from the Longevity Improvement & Fair Evidence (LIFE) Study, a research database managed by Kyushu University [12]. The study data contained medical claims data from Japan's National Health Insurance System (ages 0–74), Latter-Stage Older Persons Healthcare System (ages ≥75 and 65–74 with certain diseases), and the Public Assistance System for residents of a single municipality. In addition, we received Basic Resident Register data for this municipality. Data were accessed between August 1, 2023 and February 1, 2024. Patient identifiers were anonymized to protect privacy.

### Data cleaning and handling of missing data

No substantial missing values were found in the study variables (e.g., age, sex, comorbidities, and expenditures). The only data cleaning performed was the removal of duplicate patient ID records to ensure that each individual was included only once in the analysis.

### Study participants and setting

We constructed a study dataset by linking each participant's medical claims data and Basic Resident Register data using anonymized research IDs. From the Basic Resident Register data, we first identified residents who did not relocate from the study municipality between October 1, 2019, and March 31, 2022. This was because each municipal government serves as the insurer for residents enrolled in the target insurance systems, and individuals who move to a different municipality would be lost to follow-up. Next, we examined the claims data to identify candidate patients who received a COVID-19 diagnosis during their initial hospitalization (index hospitalization), had been treated in an ICU or emergency department (ED), and were discharged between April 1, 2020 and September 30, 2021. COVID-19 was identified using the International Classification of Diseases, 10th Revision (ICD-10) code U071. ICU and ED admissions were identified using the Japanese medical fee codes A300 and A301, which indicate claims for "Specific Intensive Care Unit Management Fees", "Emergency Care Unit Admission Fees", and "High Care Unit Admission/Medical Management Fees".

### Exclusion criteria

We first excluded individuals who fulfilled any of the following criteria: (i) previous hospitalization within 6 months before the index hospitalization, (ii) duplicate patient ID records in the study data, and (iii) age < 18 years. To analyze the primary outcome of mortality, we also excluded (iv) patients who died during hospitalization or on the discharge date and (v) patients with a length of stay (LOS) of only 1 day. To analyze the secondary outcome of total medical expenditures, we further excluded (vi) patients who died within 180 days post-discharge to specifically assess the ongoing costs for survivors and avoid the distinct expenditure patterns associated with end-of-life care [13] and (vii) patients with no medical claims data within 180 days post-discharge. As our study cohort was limited to persons who resided within the study municipality throughout the follow-up period, an absence of claims data would signify that these individuals did not use any insured medical services. The exclusion of these zero-expenditure patients allowed us to focus the analysis on the distribution and determinants of expenditures among patients who used healthcare services after discharge.

### Study exposure

The study exposure was the use of either MV with endotracheal intubation (medical fee codes: J044 and J045) or ECMO (K916) during the index hospitalization. Patients who received either of these interventions were categorized as the MV/ECMO group, and all other patients were categorized as the non-MV/ECMO group. For this study, the MV/ECMO group was used to represent severe COVID-19 patients.

## Study outcomes

The primary outcome was 180-day post-discharge mortality, and the secondary outcome was 180-day post-discharge total medical expenditures (encompassing all insurance-covered inpatient, outpatient, pharmacy, and dental costs). We set the 180-day post-discharge period for this analysis because COVID-19 healthcare resource utilization has shown to peak within the first 30 days after a patient's discharge, followed by a decline over the next 6 months [9]. For expenditures, the 180-day period began from the month following discharge (excluding the discharge month). Under the Japanese healthcare system, medical claims data are collected monthly, and the hospitalization expenditures for each month are bundled together. To distinguish between the expenditures for the index hospitalization and new expenditures incurred independently in the post-discharge period, we did not include expenditures in the discharge month.

## Covariates

To identify potential risk factors for the outcome, we included the following covariates in our analytical models: age on admission, sex, obesity, LOS, hospitalization expenditure, continuous hemodiafiltration (medical fee code: J038), delirium on admission (ICD-10 codes: F059, F050, F051) [14], and comorbidities. Information on comorbidities was extracted as ICD-10 codes from each patient's medical claims during the 180-day period before hospitalization [15]. The presence of a comorbidity was defined by at least one relevant ICD-10 code appearing in an inpatient or outpatient claims record during this look-back period. The following comorbidities were analyzed: hypertension (ICD-10 code: I10), diabetes (E10–E14), lower respiratory disease (J40–J47), heart disease (I20–I25, I30–I52) [16], kidney disease (N18, I12), cerebrovascular disease (I60–I69), dementia (F01–F03, G31.0, G31.8), cancer (C00–D49), and liver disease (K70, K72–K74, K76.0, K76.1, K76.6, K76.8). Information on obesity was extracted from the Elixhauser Comorbidity Index [17].

## Statistical analysis

The participants' baseline characteristics were descriptively summarized using the median and interquartile range (IQR) for continuous variables and numbers with percentages for categorical variables. All expenditures were converted from Japanese yen to US dollars at an exchange rate of ¥143.30 = $1.00 (as of August 2023).

For the primary outcome, the $\chi^2$ test was used to assess the difference in 180-day post-discharge mortality (binary variable) between the MV/ECMO and non-MV/ECMO groups. Mortality was then visualized using cumulative incidence curves, which were compared using the log-rank test. A Cox regression analysis was performed to calculate the adjusted hazard ratios (HRs) and 95% confidence intervals (CIs) of the MV/ECMO group (reference: non-MV/ECMO group) for mortality [18]. The HRs and 95% CIs of the covariates were also calculated.

For the secondary outcome, the Mann–Whitney $U$ test was used to assess the difference in 180-day post-discharge total medical expenditures (continuous variable) between the MV/ECMO and non-MV/ECMO groups. A generalized linear model [19] with a gamma distribution and log-link function was used to evaluate the association of the MV/ECMO group (reference: non-MV/ECMO group) with expenditures. The exponentiated coefficients and 95% CIs were calculated for both the exposure and covariates.

In addition, the number of hospital readmissions and the components of total medical expenditures (inpatient, outpatient, pharmacy, and dental) were evaluated as secondary analyses. The median number of hospital readmissions and costs for each component were compared between the groups using the Mann–Whitney $U$ test.

The significance threshold was set at $p < 0.05$ (two-tailed), and all analyses were performed using R statistical software version 4.0.2 (R Foundation for Statistical Computing, Vienna, Austria).

## Sensitivity analyses

First, we addressed the possibility of survivorship bias by conducting a sensitivity analysis of total medical expenditures in a cohort that also included the 461 patients who died within 180 days post-discharge. In this cohort, we

compared the median total medical expenditures between the MV/ECMO and non-MV/ECMO groups using the Mann–Whitney $U$ test.

Second, we stratified our cohort into three age groups (<65 years, 65–74 years, and ≥75 years), and re-performed the primary analyses for mortality and expenditures with these age groups instead of age as a continuous variable. The age groups were set based on a previous study of older adults with COVID-19 in Japan [20].

Third, to assess the impact of comorbidity burden, Charlson Comorbidity Index and Elixhauser Comorbidity Index scores (i.e., composite measures) were calculated for each patient and included as covariates instead of individual conditions in the analytical models.

Fourth, to assess the potential effects of different COVID-19 variant periods, the study cohort was stratified into two groups according to patient admission date: the pre-Delta variant period (April 1, 2020, to June 30, 2021) and the Delta variant period (July 1, 2021, to September 30, 2021). These periods were based on the documented pandemic timeline in Japan [21]. To assess whether the impact of MV/ECMO use on the outcomes was modified by the variant period, an interaction term between the Delta variant period and MV/ECMO was included in the analytical models. Additionally, we performed a separate exploratory analysis comparing the MV-only group and the ECMO group (which may include MV) to explore potential differences in outcomes related to treatment intensity. These results are provided as Supplementary Table S9 in **S1 File**.

### Ethics statement

This study was conducted in accordance with the principles of the Declaration of Helsinki. The study was approved by the Kyushu University Institutional Review Board for Clinical Research (Approval no. 22114−03). The review board waived the requirement for informed consent due to the study's retrospective nature and because all records were de-identified and fully anonymized before being received by the authors.

## Results

The patient selection process is presented in Fig 1. We identified 4,532 eligible COVID-19 patients. After applying the exclusion criteria, the analysis of 180-day post-discharge mortality (primary outcome) was conducted using 3,938 patients (MV/ECMO group: 493, non-MV/ECMO group: 3,445), while the analysis of 180-day post-discharge total medical expenditures (secondary outcome) was conducted using 3,407 patients (MV/ECMO group: 412, non-MV/ECMO group: 2,995).

### Participants' baseline characteristics

Table 1 summarizes the characteristics of the 4,532 COVID-19 patients in the MV/ECMO and non-MV/ECMO groups. Among these patients, the MV/ECMO group had a longer LOS (median: 19 days, IQR: 8–37 days) than the non-MV/ECMO group (15 days, 9–25 days). We observed other intergroup differences in in-hospital mortality (35% vs. 7.7%), median hospital expenditures ($18,239 vs. $11,204), and delirium on admission (9.5% vs. 8.4%). There were also differences in comorbidities, such as a higher prevalence of heart disease in the MV/ECMO group than the non-MV/ECMO group (59% vs. 32%).

### Main outcomes

Table 2 presents the results of the univariate comparisons of outcomes between the patient groups. The $\chi^2$ test showed that the MV/ECMO group had a significantly higher incidence of 180-day post-discharge mortality than the non-MV/ECMO group (16.0% vs. 11.1%, $p = 0.002$). Similarly, the Mann–Whitney $U$ test showed that the MV/ECMO group had significantly higher 180-day post-discharge total medical expenditures than the non-MV/ECMO group ($8,732 vs. $3,460, $p < 0.001$).

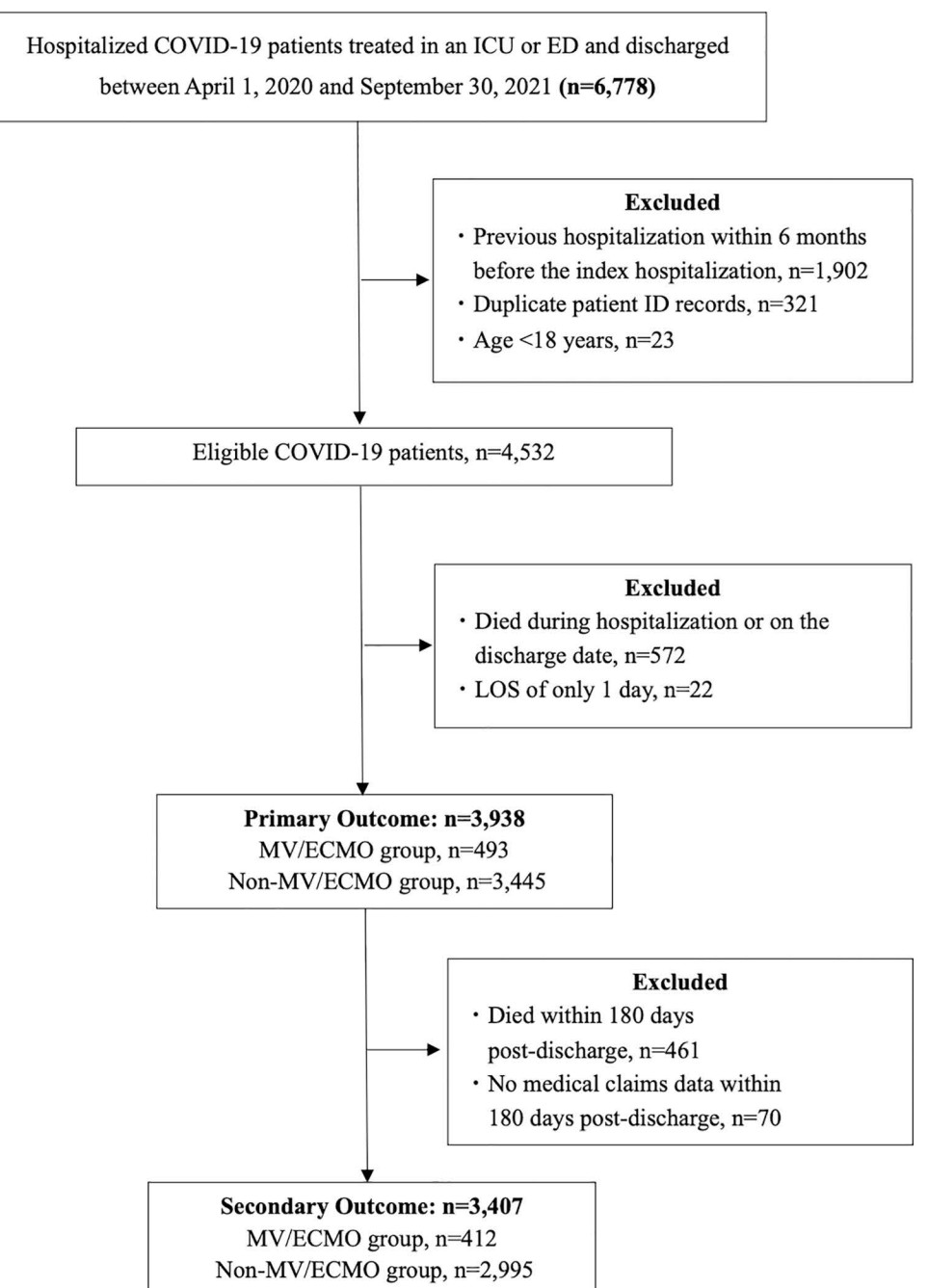

**Fig 1. Flowchart of the Patient Selection Process.** ECMO, extracorporeal membrane oxygenation; ED, emergency department; ICU, intensive care unit; LOS, length of stay; MV, mechanical ventilation.

The cumulative incidence curves of mortality were compared between the MV/ECMO and non-MV/ECMO groups (Fig 2). The MV/ECMO group had a significantly higher probability of death than the non-MV/ECMO group (log-rank $p < 0.001$).

**Table 1. Baseline Characteristics of Hospitalized COVID-19 Patients According to MV/ECMO Use.**

| Characteristics | Eligible COVID-19 Patients | | Primary Outcome Analysis | | Secondary Outcome Analysis | |
|---|---|---|---|---|---|---|
| | MV/ECMO | Non-MV/ECMO | MV/ECMO | Non-MV/ECMO | MV/ECMO | Non-MV/ECMO |
| n | n=771 | n=3,761 | n=493 | n=3,445 | n=412 | n=2,995 |
| Age on admission, years, median [IQR] | 77 [70, 84] | 78 [70, 85] | 76 [69, 83] | 78 [70, 85] | 76 [69, 82] | 77 [70, 84] |
| Sex, n (%): | | | | | | |
| Male | 439 (57%) | 1,945 (52%) | 285 (58%) | 1,781 (52%) | 233 (57%) | 1,544 (52%) |
| Obesity, n (%) | 10 (1.3%) | 41 (1.1%) | 8 (1.6%) | 40 (1.2%) | 6 (1.5%) | 37 (1.2%) |
| Insurance type: | | | | | | |
| National Health Insurance | 257 (33.3%) | 1,241 (33%) | 180 (36.5%) | 1,191 (34.6%) | 154 (37.4%) | 1,086 (36.2%) |
| Public Assistance System | 32 (4.2%) | 125 (3.3%) | 23 (4.7%) | 119 (3.5%) | 21 (5.1%) | 95 (3.2%) |
| Latter-Stage Older Persons Health-care System | 482 (62.5%) | 2,395 (63.7%) | 290 (58.8%) | 2,135 (61.9%) | 237 (57.5%) | 1,814 (60.6%) |
| LOS, days, median [IQR] | 19 [8, 37] | 15 [9, 25] | 26 [16, 44] | 15 [9, 26] | 25 [16, 42] | 15 [9, 25] |
| Hospitalization expenditure, USD, median [IQR] | 18,239 [8,242, 33,724] | 11,204 [6,910, 17,862] | 23,190 [13,254, 38,274] | 11,406 [7,156, 17,955] | 23,322 [13,367, 38,441] | 11,400 [7,145, 17,723] |
| In-hospital death | 273 (35%) | 289 (7.7%) | 0 | 0 | 0 | 0 |
| CHDF, n (%) | 84 (11%) | 77 (2.0%) | 41 (8.3%) | 64 (1.9%) | 29 (7.0%) | 52 (1.7%) |
| Delirium on admission, n (%) | 73 (9.5%) | 317 (8.4%) | 60 (12%) | 281 (8.2%) | 47 (11%) | 242 (8.1%) |
| Hypertension, n (%) | 248 (32%) | 1,349 (36%) | 198 (40%) | 1,249 (36%) | 173 (42%) | 1,105 (37%) |
| Diabetes, n (%) | 179 (23%) | 858 (23%) | 137 (28%) | 778 (23%) | 117 (28%) | 681 (23%) |
| Lower respiratory disease, n (%) | 67 (8.7%) | 274 (7.3%) | 51 (10%) | 253 (7.3%) | 37 (9.0%) | 207 (6.9%) |
| Heart disease, n (%) | 453 (59%) | 1,187 (32%) | 268 (54%) | 1,061 (31%) | 224 (54%) | 917 (31%) |
| Kidney disease, n (%) | 60 (7.8%) | 178 (4.7%) | 35 (7.1%) | 149 (4.3%) | 28 (6.8%) | 123 (4.1%) |
| Cerebrovascular disease, n (%) | 151 (20%) | 889 (24%) | 109 (22%) | 828 (24%) | 95 (23%) | 721 (24%) |
| Dementia, n (%) | 41 (5.3%) | 256 (6.8%) | 30 (6.1%) | 221 (6.4%) | 24 (5.8%) | 172 (5.7%) |
| Cancer, n (%) | 115 (15%) | 779 (21%) | 76 (15%) | 742 (22%) | 57 (14%) | 613 (20%) |
| Liver disease, n (%) | 20 (2.6%) | 83 (2.2%) | 9 (1.8%) | 77 (2.2%) | 6 (1.5%) | 62 (2.1%) |
| Charlson score on admission, median [IQR] | 2.0 [1.0, 3.0] | 2.0 [1.0, 3.0] | 2.0 [1.0, 3.0] | 2.0 [1.0, 3.0] | 2.0 [1.0, 3.0] | 2.0 [1.0, 3.0] |
| Elixhauser score on admission, median [IQR] | 3.0 [2.0, 5.0] | 3.0 [2.0, 4.0] | 3.0 [2.0, 5.0] | 3.0 [1.0, 4.0] | 3.0 [2.0, 5.0] | 3.0 [1.0, 4.0] |

CHDF, continuous hemodiafiltration; ECMO, extracorporeal membrane oxygenation; IQR, interquartile range; LOS, length of stay; MV, mechanical ventilation.

Table 3 summarizes the results of the Cox regression analysis of 180-day post-discharge mortality. The MV/ECMO group showed a significantly higher hazard for mortality than the non-MV/ECMO group (HR: 1.66, 95% CI: 1.27–2.15). The following covariates were also significantly associated with mortality: age (HR: 1.06, 95% CI: 1.05–1.07), men (1.24, 1.02–1.50), LOS (1.01, 1.01–1.02), hospitalization expenditure (1.00, 1.00–1.00), hypertension (0.79, 0.64–0.96), lower respiratory disease (1.43, 1.06–1.92), dementia (1.48, 1.10–1.99), cancer (1.92, 1.56–2.36), and liver disease (1.93, 1.19–3.13).

The results of the generalized linear model analysis of 180-day post-discharge total medical expenditures are presented in Table 4. The MV/ECMO group showed significantly higher expenditures than the non-MV/ECMO group (Exp[β]: 1.49, 95% CI: 1.29–1.73). The following covariates were also significantly associated with expenditures: age (Exp[β]: 1.02, 95% CI: 1.01–1.02), LOS (1.01, 1.00–1.01), lower respiratory disease (1.19, 1.01–1.43), kidney disease (1.60, 1.29–2.01), cerebrovascular disease (1.74, 1.56–1.94), and cancer (1.28, 1.14–1.44).

**Table 2. Comparisons of 180-Day Post-Discharge Mortality and Total Medical Expenditures in Hospitalized COVID-19 Patients According to MV/ECMO Use.**

| | Primary Outcome Analysis | | | |
| --- | --- | --- | --- | --- |
| | MV/ECMO | Non-MV/ECMO | *p*-value | Statistical test |
| 180-day post-discharge mortality, n (%) | n = 493 | n = 3,445 | | |
| | 79 (16.0) | 382 (11.1) | 0.002 | χ² test |
| | Secondary Outcome Analysis | | | |
| | MV/ECMO | Non-MV/ECMO | *p*-value | Statistical test |
| 180-day post-discharge total medical expenditures, USD, median [IQR] | n = 412 | n = 2,995 | | |
| | 8,732 [2,325, 25,023] | 3,460 [1,255, 13,583] | <0.001 | Mann–Whitney *U* test |

ECMO, extracorporeal membrane oxygenation; IQR, interquartile range; MV, mechanical ventilation.

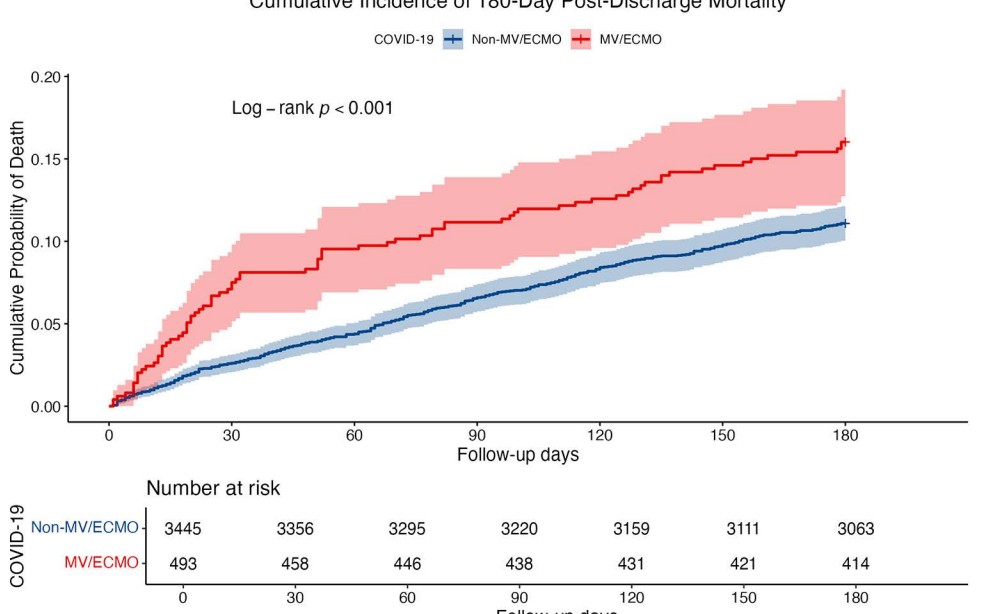

**Fig 2. Cumulative Incidence Curves of 180-Day Post-Discharge Mortality.** ECMO, extracorporeal membrane oxygenation; MV, mechanical ventilation.

## Secondary analyses: readmission and cost components

The median number of hospital readmissions within 180 days post-discharge was significantly higher in the MV/ECMO group than in the non-MV/ECMO group (1.0 [IQR: 0.0, 1.0] vs. 0.0 [IQR: 0.0, 1.0], *p* < 0.001). Next, the analysis of cost components showed that median inpatient costs were significantly higher in the MV/ECMO group than in the non-MV/ECMO group ($671 vs. $359, *p* < 0.001). In contrast, there were no significant intergroup differences in outpatient costs ($69 vs. $67, *p* = 0.607), pharmacy costs ($45 vs. $40, *p* = 0.393), or dental costs ($0 vs. $0, *p* = 0.313).

## Sensitivity analyses

First, we compared the median total medical expenditures between the MV/ECMO and non-MV/ECMO groups in a cohort that included patients who died within 180 days post-discharge (n = 3,868). The median expenditures of the MV/ECMO

**Table 3. Cox Regression Analysis of 180-Day Post-Discharge Mortality (n = 3,938).**

| Independent Variables | Hazard Ratio | 95% CI | p-value |
|---|---|---|---|
| MV/ECMO (ref: Non-MV/ECMO) | 1.66 | 1.27–2.15 | <0.001 |
| Age | 1.06 | 1.05–1.07 | <0.001 |
| Male (ref: Female) | 1.24 | 1.02–1.50 | 0.028 |
| Obesity | 0.83 | 0.26–2.60 | 0.74 |
| LOS | 1.01 | 1.01–1.02 | <0.001 |
| Hospitalization expenditure | 1.00 | 1.00–1.00 | 0.025 |
| Delirium on admission | 0.96 | 0.71–1.29 | 0.77 |
| Hypertension | 0.79 | 0.64–0.96 | 0.018 |
| Diabetes | 1.14 | 0.91–1.41 | 0.25 |
| Lower respiratory disease | 1.43 | 1.06–1.92 | 0.019 |
| Heart disease | 1.01 | 0.83–1.23 | 0.95 |
| Kidney disease | 1.33 | 0.92–1.93 | 0.13 |
| Cerebrovascular disease | 1.00 | 0.80–1.25 | 0.97 |
| Dementia | 1.48 | 1.10–1.99 | 0.01 |
| Cancer | 1.92 | 1.56–2.36 | <0.001 |
| Liver disease | 1.93 | 1.19–3.13 | 0.007 |

Concordance = 0.717 (standard error = 0.011). CI, confidence interval; ECMO, extracorporeal membrane oxygenation; LOS, length of stay; MV, mechanical ventilation.

**Table 4. Generalized Linear Model Analysis of 180-Day Post-Discharge Total Medical Expenditures (n = 3,407).**

| Independent Variables | Exp(β) | 95% CI | p-value |
|---|---|---|---|
| MV/ECMO (ref: Non-MV/ECMO) | 1.49 | 1.29–1.73 | <0.001 |
| Age | 1.02 | 1.01–1.02 | <0.001 |
| Male (ref: Female) | 1.04 | 0.95–1.14 | 0.38 |
| Obesity | 1.01 | 1.00–1.01 | 0.053 |
| LOS | 1.01 | 1.00–1.01 | <0.001 |
| Hospitalization expenditure | 1.00 | 1.00–1.00 | 0.28 |
| Delirium on admission | 1.08 | 0.92–1.28 | 0.34 |
| Hypertension | 1.03 | 0.93–1.13 | 0.61 |
| Diabetes | 1.04 | 0.93–1.16 | 0.48 |
| Lower respiratory disease | 1.19 | 1.01–1.43 | 0.046 |
| Heart disease | 1.01 | 0.91–1.12 | 0.86 |
| Kidney disease | 1.60 | 1.29–2.01 | <0.001 |
| Cerebrovascular disease | 1.74 | 1.56–1.94 | <0.001 |
| Dementia | 1.17 | 0.97–1.42 | 0.12 |
| Cancer | 1.28 | 1.14–1.44 | <0.001 |
| Liver disease | 1.23 | 0.91–1.73 | 0.2 |

CI, confidence interval; ECMO, extracorporeal membrane oxygenation; LOS, length of stay; MV, mechanical ventilation.

group were 2.2 times higher than that of the non-MV/ECMO group ($8,285 [IQR: $2,054–$23,628] vs. $3,792 [IQR: $1,256–$14,160], $p < 0.001$).

Second, a Cox regression analysis was performed to examine the associations of the age groups with mortality (S1 Table in S1 File). After adjusting for the other covariates, older age was found to be a strong predictor of mortality. Compared to patients aged <65 years, the hazard for mortality was significantly higher for patients aged 65–74 years (HR: 1.83, 95% CI: 1.15–2.90), and was even more pronounced for those aged ≥75 years (3.21, 2.10–4.90). Similarly, a generalized linear model was used to examine the associations of the age groups with total medical expenditures (S2 Table in S1 File). Compared to patients aged <65 years, expenditures were significantly higher for those aged 65–74 years (Exp[β]: 1.37, 95% CI: 1.18–1.58) and ≥75 years (1.53, 1.34–1.74). In both analyses, MV/ECMO use remained significantly associated with the outcomes.

Third, we assessed the alternate impact of comorbidities on the outcomes by using the Charlson Comorbidity Index (S3–S4 Tables in S1 File) and the Elixhauser Comorbidity Index (S5–S6 Tables in S1 File) in place of individual conditions in the analytical models. In the Cox regression models, both the Charlson score (HR: 1.10, 95% CI: 1.03–1.17) and Elixhauser score (1.07, 1.02–1.12) were positively associated with mortality. Similarly, the generalized linear models showed that both the Charlson score (Exp[β]: 1.12, 95% CI: 1.08–1.16) and Elixhauser score (1.07, 1.05–1.10) were positively associated with total medical expenditures. MV/ECMO use remained significantly associated with the outcomes for both comorbidity indices.

Fourth, we assessed the potential influence of the COVID-19 Delta and pre-Delta variant periods. Neither the Delta variant period itself ($p = 0.38$) nor the interaction term between the Delta variant period and MV/ECMO ($p = 0.85$) were significantly associated with mortality (S7 Table in S1 File), suggesting that the association between MV/ECMO and mortality did not significantly differ between the variant periods. In contrast, the interaction term between the Delta variant period and MV/ECMO was significantly associated with total medical expenditures ($p = 0.02$) (S8 Table in S1 File). While MV/ECMO was associated with 1.38-times higher expenditures in the pre-Delta variant period, this effect was magnified by 1.56 times during the Delta variant period. Even when accounting for COVID-19 variant periods, MV/ECMO use remained significantly associated with both outcomes.

## Discussion

Our study demonstrated that severe COVID-19 patients requiring MV or ECMO had significantly higher 180-day post-discharge mortality and medical expenditures compared to those who did not require such interventions. Comorbidities including dementia, cancer, kidney disease, and cerebrovascular disease were associated with poorer outcomes and increased costs. These findings were robust across multiple sensitivity analyses and align with previous studies reporting increased mortality among patients with severe COVID-19 [6–8] and increased healthcare utilization following COVID-19 infection [9]. Our cost component analysis revealed that the greater financial burden on patients in the MV/ECMO group was primarily driven by subsequent hospitalization costs, rather than by an increased use of outpatient, pharmacy, or dental services.

In addition to our main findings, the sensitivity analyses consistently demonstrated that older age and higher comorbidity burden are key determinants of both outcomes. The age-stratified analysis revealed that patients aged ≥65 years had significantly higher mortality and medical expenditures than younger patients, which underscores the vulnerability of this population and corroborates the findings of a previous Japanese study [20]. Furthermore, the robustness of the associations between comorbidities and the outcomes was confirmed, as both the Charlson and Elixhauser scores were significant predictors of mortality and expenditures. Our results are consistent with a large-scale study from the US that identified older age and comorbidities as primary drivers of healthcare resource utilization and costs in COVID-19 patients [22].

The interaction term between the Delta variant period and MV/ECMO was significantly associated with total medical expenditures, but not with mortality. This discrepancy may be explained by the clinical characteristics of the SARS-CoV-2

Delta variant, which was associated with more severe acute illness and a higher need for intensive care than previous variants [23]. This higher severity may also have led to a greater burden of long-term complications in survivors. A US study reported that infection during the Delta variant period was associated with a higher risk of post-acute sequelae (e.g., pulmonary and neurological disorders) than earlier variants [24]. Therefore, an increased incidence of chronic non-fatal health problems among survivors of severe Delta variant COVID-19 provides a strong potential explanation for our findings.

Our results from a Japanese municipality were similar to those of studies from other countries. In a Swedish investigation of ICU-admitted COVID-19 patients, MV users were found to be significantly associated with higher 180-day post-discharge mortality than non-MV users (35.7% vs. 19.1%, $p < 0.001$) [6]. Because MV-related complications, such as ventilator-associated pneumonia, are associated with increased mortality among severe COVID-19 patients [25], early weaning from MV may help to reduce mortality risk.

An exploratory comparison showed that ECMO survivors had lower post-discharge mortality and medical expenditures than MV-only patients (Supplementary Table S9 in S1 File). This finding may reflect survivor selection. Because the analysis was limited to patients who survived to hospital discharge, the ECMO group may represent a selected population with better recovery potential. Therefore, this result should be interpreted with caution.

Next, our present study also detected an association between dementia and 180-day post-discharge mortality. A US study similarly reported an association between dementia and increased mortality in COVID-19 patients (odds ratio: 1.29, 95% CI: 1.07–1.56) [26]. In addition, a nationwide cohort study in South Korea showed that patients with Alzheimer's disease were more susceptible to severe complications and mortality related to COVID-19 (odds ratio: 3.09, 95% CI: 1.54–3.28) [27]. The association between dementia and mortality risk may be explained by delays in seeking treatment for COVID-19 (potentially resulting from impaired symptom recognition) that can exacerbate disease progression [28,29].

Furthermore, a US cohort study on cancer patients who developed COVID-19 reported that the mortality rate of ICU-admitted patients was three times higher than that of ward-admitted patients [30]. Cancer treatments such as chemotherapy and radiotherapy can impair immune function, leading to more severe COVID-19 and prolonged recovery that increases mortality risk and medical expenditures. Our study also found that liver disease as a comorbidity was associated with 180-day post-discharge mortality. This was supported by the findings of a multi-center study conducted in the US, which noted that liver disease increased mortality risk in COVID-19 patients (HR: 2.8, 95% CI: 1.9–4.0) [31].

The MV/ECMO group in our study required significantly more healthcare resources than the non-MV/ECMO group, with approximately 2.5-times higher medical expenditures over the follow-up period. A US cohort study reported that a COVID-19 diagnosis was associated with a 1.6-fold increase in healthcare visits and a 2.7-fold increase in monthly medical expenditures for 6 months after diagnosis [9]. Our study also showed that LOS in the index hospitalization was associated with higher post-discharge medical expenditures. A multi-regional cohort study conducted in Japan similarly found that LOS was positively associated with 1-year post-discharge medical expenditures [32].

Longer LOS may indicate more severe disease during the index hospitalization, which could drive increases in subsequent expenditures through the need for additional post-discharge care. Moreover, kidney disease and cerebrovascular disease were also linked to higher medical expenditures, potentially due to the need for dialysis or rehabilitation therapy [33,34]. As COVID-19 can exacerbate cardiovascular disease [35], physicians should not only focus on respiratory diseases, but also consider these underlying conditions as important risk factors when treating patients with severe COVID-19. From a health economic perspective, the early identification and treatment of patients at risk of severe COVID-19 may help to reduce their subsequent expenditures, thereby reducing the financial burden on both patients and insurers.

## Limitations

This study has several limitations. First, this was an observational study conducted in a single Japanese municipality. Therefore, the analyses do not prove causation between severe COVID-19 and the study outcomes, and our findings may

have limited generalizability. Second, COVID-19 severity was inferred solely from claims data, potentially leading to mis-classifications due to coding errors or incomplete claims. These misclassifications may also have affected the identification of comorbidities. In addition, the claims data did not include detailed clinical information, such as laboratory data (e.g., C-reactive protein) and severity markers (e.g., oxygen saturation and APACHE II scores). The absence of these variables may have led to residual confounding, potentially distorting our results. Third, we could not account for variations in clinical management across hospitals. Fourth, medical expenditures for comorbidities that were present before the index hospitalization were not evaluated. Finally, our analysis of expenditures excluded patients who died within 180 days post-discharge, which may have introduced survivorship bias. Although a sensitivity analysis confirmed the robustness of our results, we could not account for death as a competing risk for cost estimation. To build upon our findings, future research should aim to address these limitations. Specifically, prospective multi-center cohort studies incorporating detailed clinical data and patient-reported outcomes would provide a more comprehensive understanding of the long-term trajectory of severe COVID-19 survivors.

## Conclusion

COVID-19 patients requiring MV or ECMO experience high post-discharge mortality and incur substantial medical expenditures. Comorbidities such as dementia, cancer, kidney disease, and cerebrovascular disease contribute to these adverse outcomes, underscoring the need for targeted post-discharge care and resource allocation.

## Supporting information

**S1 File. S1 Table.** Cox Regression Analysis of 180-Day Post-Discharge Mortality with Age Groups. Concordance = 0.718 (standard error = 0.011). CI, confidence interval; ECMO, extracorporeal membrane oxygenation; LOS, length of stay; MV, mechanical ventilation. **S2 Table.** Generalized Linear Model Analysis of 180-Day Post-Discharge Total Medical Expenditures with Age Groups. CI, confidence interval; ECMO, extracorporeal membrane oxygenation; LOS, length of stay; MV, mechanical ventilation. **S3 Table.** Cox Regression Analysis of 180-Day Post-Discharge Mortality with Charlson Comorbidity Index Scores. Concordance = 0.70 (standard error = 0.012). CI, confidence interval; ECMO, extracorporeal membrane oxygenation; LOS, length of stay; MV, mechanical ventilation. **S4 Table.** Cox Regression Analysis of 180-Day Post-Discharge Mortality with Elixhauser Comorbidity Index Scores. Concordance = 0.70 (standard error = 0.012). CI, confidence interval; ECMO, extracorporeal membrane oxygenation; LOS, length of stay; MV, mechanical ventilation. **S5 Table.** Generalized Linear Model Analysis of 180-Day Post-Discharge Total Medical Expenditures with Charlson Comorbidity Index Scores. CI, confidence interval; ECMO, extracorporeal membrane oxygenation; LOS, length of stay; MV, mechanical ventilation. **S6 Table.** Generalized Linear Model Analysis of 180-Day Post-Discharge Total Medical Expenditures with Elixhauser Comorbidity Index Scores. CI, confidence interval; ECMO, extracorporeal membrane oxygenation; LOS, length of stay; MV, mechanical ventilation. **S7 Table.** Cox Regression Analysis of 180-Day Post-Discharge Mortality with COVID-19 Variant Periods. Concordance = 0.718 (standard error = 0.011). CI, confidence interval; ECMO, extracorporeal membrane oxygenation; LOS, length of stay; MV, mechanical ventilation. **S8 Table.** Generalized Linear Model Analysis of 180-Day Post-Discharge Total Medical Expenditures with COVID-19 Variant Periods. CI, confidence interval; ECMO, extracorporeal membrane oxygenation; LOS, length of stay; MV, mechanical ventilation. **S9 Table.** Comparisons of 180-Day Post-Discharge Mortality and Total Medical Expenditures Between MV-Only and ECMO Patients. ECMO, extracorporeal membrane oxygenation; IQR, interquartile range; MV, mechanical ventilation.
(ZIP)

## Acknowledgments

We are grateful to the staff of the participating municipalities for providing the data used in this study.

## Author contributions

**Conceptualization:** Jun Kawabata, Haruhisa Fukuda.

**Data curation:** Megumi Maeda, Haruhisa Fukuda.

**Formal analysis:** Jun Kawabata, Kenichi Goto, Haruhisa Fukuda.

**Funding acquisition:** Haruhisa Fukuda.

**Investigation:** Jun Kawabata, Haruhisa Fukuda.

**Methodology:** Jun Kawabata, Kenichi Goto, Haruhisa Fukuda.

**Project administration:** Jun Kawabata, Haruhisa Fukuda.

**Resources:** Megumi Maeda, Haruhisa Fukuda.

**Supervision:** Kenichi Goto, Megumi Maeda, Haruhisa Fukuda.

**Visualization:** Jun Kawabata.

**Writing – original draft:** Jun Kawabata, Kenichi Goto.

**Writing – review & editing:** Jun Kawabata, Kenichi Goto, Haruhisa Fukuda.

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
