## [Decision Letter · Decision Letter 0]

10 Jul 2025

Dear Dr. Kenichi,

Thank you for submitting your manuscript to PLOS ONE. After careful consideration, we feel that it has merit but does not fully meet PLOS ONE’s publication criteria as it currently stands. Therefore, we invite you to submit a revised version of the manuscript that addresses the points raised during the review process.

We look forward to receiving your revised manuscript.

Kind regards,

Emma Campbell, Ph.D

Staff Editor

PLOS ONE

Journal Requirements:

https://journals.plos.org/plosone/s/file?id=ba62/PLOSOne_formatting_sample_title_authors_affiliations.pdf ..

3. For studies involving third-party data, we encourage authors to share any data specific to their analyses that they can legally distribute. PLOS recognizes, however, that authors may be using third-party data they do not have the rights to share. When third-party data cannot be publicly shared, authors must provide all information necessary for interested researchers to apply to gain access to the data. (https://journals.plos.org/plosone/s/data-availability#loc-acceptable-data-access-restrictions)

5. Please upload a copy of supplementary material which you refer to in your text on page 7.

Reviewers' comments:

Reviewer's Responses to Questions

**Comments to the Author**

1. Is the manuscript technically sound, and do the data support the conclusions?

Reviewer #1: Yes

2. Has the statistical analysis been performed appropriately and rigorously?

Reviewer #1: Yes

3. Have the authors made all data underlying the findings in their manuscript fully available?

Reviewer #1: Yes

4. Is the manuscript presented in an intelligible fashion and written in standard English?

Reviewer #1: Yes

Reviewer #1: Overall recommendation:

-While the conclusions are supported by the data, the manuscript should emphasize that associations do not prove causation due to observational design.

-Consider discussing potential residual confounding and unmeasured severity indicators.

-Future studies with clinical data and multi-center cohorts would strengthen evidence.

Abstract:

The phrase "Outcomes of health condition and economic of patients" is awkward and unclear.

Suggestion: Outcomes related to health status and economic burden among patients who experienced critical COVID-19 remain insufficiently studied.

Use consistent terminology (e.g., "mechanical ventilation" vs. "ventilator use").

Avoid redundancy (e.g., "COVID-19 patients who did or did not require mechanical ventilation in ICU or ED" can be simplified).

Introduction:

Some sentences are verbose or repetitive.

Use consistent terminology (e.g., "severe COVID-19" rather than "severe cases of COVID-19").

Clarify ambiguous phrases (e.g., "patients with severe COVID-19 who are discharged from an ICU may continue to have physical, mental, and cognitive symptoms" — specify "post-discharge").

Example for first paragraph:

The COVID-19 pandemic has led to a surge of critically ill patients requiring intensive care worldwide [1]. Patients discharged from ICUs after severe COVID-19 often experience persistent physical, mental, and cognitive impairments [2,3], and face increased risks of poor prognosis and higher healthcare costs post-discharge [4,5]

Methods:

-Clarify some complex sentences for readability.

-Use consistent terminology for insurance systems.

-Avoid passive voice where possible.

-Clarify exclusion criteria and study population flow.

-Correct minor grammar issues (e.g., "We have accessed between August 1, 2023, and February 1, 2024" → "Data were accessed between...").

Example revision snippet:

This cohort study used data from the LIFE Study, a research database managed by Kyushu University, containing claims data from Japan's National Health Insurance System (ages 0–74) and the Latter-Stage Elderly Healthcare System (ages ≥75 and 65–74 with certain diseases) for residents of a single municipality [11]. Data were accessed between August 1, 2023, and February 1, 2024. Patient identifiers were anonymized to protect privacy.

-Severity of COVID-19 is inferred indirectly via mechanical ventilation/ECMO use and ICU billing codes, which may not fully capture clinical severity or differences in care.

Suggestion: Consider incorporating additional clinical severity indicators if available (e.g., oxygen saturation, laboratory markers, APACHE II scores).

If unavailable, explicitly acknowledge this limitation and discuss potential residual confounding.

if possible - Use propensity score methods or inverse probability weighting to better balance groups on observed covariates and reduce confounding bias.

-Address Potential Selection Bias - Excluding patients who died during hospitalization or within 180 days post-discharge for expenditure analysis may introduce survivorship bias.

Suggestion:

Discuss this potential bias explicitly.

Consider sensitivity analyses including these patients or using methods to account for censored data (e.g., competing risks models).

Justify the exclusion of patients without claims data within 180 days (e.g., due to relocation or loss to follow-up).

Expand on Comorbidity Assessment - Comorbidities are identified using ICD-10 codes up to 180 days prior to hospitalization, but the method for defining presence (e.g., single code vs. multiple codes) is not detailed.

Suggestion:

Specify the algorithm or criteria used to define comorbidities (e.g., at least one inpatient or two outpatient claims).

Consider using validated comorbidity indices (e.g., Charlson or Elixhauser) as composite measures.

Discuss potential misclassification due to coding errors or incomplete claims.

Clarify Definition and Timing of Medical Expenditures - Medical expenditures are defined from the month following discharge through 180 days, excluding the discharge month.

Suggestion:

Provide rationale for excluding the discharge month expenditures (e.g., to separate inpatient from outpatient costs).

Clarify whether expenditures include all healthcare costs (inpatient, outpatient, pharmacy) or only specific categories.

Consider analyzing expenditures by category to identify drivers of cost differences.

If time allows and possible - Consider Additional Outcomes or Subgroup Analyses

Suggestion:

Include other relevant outcomes such as readmission rates, quality of life, or functional status if data permit.

Perform subgroup analyses by age groups, comorbidity burden, or variant periods (alpha vs. delta) to explore heterogeneity.

Explore time-varying covariates or longitudinal expenditure patterns.

Results:

-Address Missing Data and Sensitivity Analyses

The results do not mention handling of missing data or sensitivity analyses.

Suggestion:

State whether missing data were present and how they were handled.

Consider reporting sensitivity analyses if performed (e.g., including patients who died within 180 days for expenditures).

Discussion:

Avoid overly long sentences.

Clarify causal interpretations carefully.

Use consistent terminology.

Improve flow and coherence

Example revision snippet: Our study demonstrated that COVID-19 patients requiring mechanical ventilation or ECMO had significantly higher 180-day post-discharge mortality and medical expenditures compared to those who did not require such interventions. Comorbidities including dementia, cancer, kidney disease, and cerebrovascular disease were associated with worse outcomes and increased costs. These findings align with previous studies reporting increased mortality and healthcare utilization among severe COVID-19 patients.

Limitations:

Use formal language

Example: This study has several limitations. First, it was conducted in a single Japanese municipality, which may limit generalizability. Second, disease severity was inferred solely from claims data, potentially leading to misclassification. Third, variations in clinical management across facilities were not accounted for. Fourth, the impact of different SARS-CoV-2 variants (alpha, delta) on outcomes was not assessed. Additionally, pre-existing medical expenditures related to comorbidities prior to COVID-19 hospitalization were not evaluated.

Revised Conclusion:

COVID-19 patients requiring mechanical ventilation or ECMO experience high post-discharge mortality and incur substantial medical expenditures. Comorbidities such as dementia, cancer, kidney disease, and cerebrovascular disease contribute to these adverse outcomes, underscoring the need for targeted post-discharge care and resource allocation.

**Do you want your identity to be public for this peer review?** For information about this choice, including consent withdrawal, please see our For information about this choice, including consent withdrawal, please see our Privacy Policy .

Reviewer #1: **Yes:** Hardik Goswami, PhDHardik Goswami, PhD

While revising your submission, please upload your figure files to the Preflight Analysis and Conversion Engine (PACE) digital diagnostic tool, https://pacev2.apexcovantage.com/ . PACE helps ensure that figures meet PLOS requirements. To use PACE, you must first register as a user. Registration is free. Then, login and navigate to the UPLOAD tab, where you will find detailed instructions on how to use the tool. If you encounter any issues or have any questions when using PACE, please email PLOS at . PACE helps ensure that figures meet PLOS requirements. To use PACE, you must first register as a user. Registration is free. Then, login and navigate to the UPLOAD tab, where you will find detailed instructions on how to use the tool. If you encounter any issues or have any questions when using PACE, please email PLOS at figures@plos.org . Please note that Supporting Information files do not need this step.. Please note that Supporting Information files do not need this step.

---

## [Author Response · Author response to Decision Letter 1]

13 Aug 2025

Dear Editor and Reviewer,

We have addressed all comments point-by-point in the attached "Response to Reviewers" file. Thank you for your insightful comments and for providing us with the opportunity to improve our manuscript.

Sincerely,

Kenichi Goto

---

## [Decision Letter · Decision Letter 1]

12 Mar 2026

Comparison of post-discharge mortality and medical expenditures in COVID-19 patients according to mechanical ventilation and extracorporeal membrane oxygenation use: The LIFE Study

PONE-D-25-24872R1

Dear Dr. Kenichi,

We’re pleased to inform you that your manuscript has been judged scientifically suitable for publication and will be formally accepted for publication once it meets all outstanding technical requirements.

Kind regards,

Chiara Lazzeri

Academic Editor

PLOS One

Additional Editor Comments (optional):

Reviewers' comments:

Reviewer's Responses to Questions

**Comments to the Author**

Reviewer #2: (No Response)

Reviewer #3: All comments have been addressed

2. Is the manuscript technically sound, and do the data support the conclusions?

Reviewer #2: Yes

Reviewer #3: Yes

3. Has the statistical analysis been performed appropriately and rigorously?

Reviewer #2: Yes

Reviewer #3: Yes

4. Have the authors made all data underlying the findings in their manuscript fully available?

Reviewer #2: Yes

Reviewer #3: No

5. Is the manuscript presented in an intelligible fashion and written in standard English?

Reviewer #2: Yes

Reviewer #3: Yes

Reviewer #2: (No Response)

Reviewer #3: This manuscript described post-discharge mortality and healthcare expenditures among patients with severe COVID-19 in Japan. The authors have carefully reflected on the previous reviewers' insights, leading to significant refinements in both the methodological framework and the overall substance of the paper. I checked that the revised manuscript has been adjusted to satisfy the reviewers' requirements. This study is well-structured and has clear research questions, but I have a few recommendations.

1. According to the exclusion criteria, those who died within 180 days or had no claims data were excluded.

“The exclusion of these zero-expenditure patients allowed for a more accurate analysis of the distribution and determinants of expenditures by focusing on the population that used healthcare services post-discharge.” In more accurate analysis the expression is too strong.

I recommend slightly relaxing this requirement by “excluding patients with zero expenditures, allowing us to focus the analysis on the distribution and determinants of expenditures among patients who used healthcare services after discharge”

2. I would like to know if there's a separate analysis of the group that used MV only. Since the severity of patients on ECMO is typically higher and there may be differences compared to the group that used only MV, I would appreciate it if you could add this analysis as supplementary data

**Do you want your identity to be public for this peer review?** For information about this choice, including consent withdrawal, please see our For information about this choice, including consent withdrawal, please see our Privacy Policy .

Reviewer #2: No

Reviewer #3: **Yes:** Su Hwan LeeSu Hwan Lee

---

## [Editor Report · Acceptance letter]

PONE-D-25-24872R1

PLOS One

Dear Dr. Goto,

I'm pleased to inform you that your manuscript has been deemed suitable for publication in PLOS One. Congratulations! Your manuscript is now being handed over to our production team.

Kind regards,

on behalf of

Dr. Chiara Lazzeri

Academic Editor

PLOS One